# The Effect of Sertoli Cells on Xenotransplantation and Allotransplantation of Ventral Mesencephalic Tissue in a Rat Model of Parkinson’s Disease

**DOI:** 10.3390/cells8111420

**Published:** 2019-11-11

**Authors:** Yun-Ting Jhao, Chuang-Hsin Chiu, Chien-Fu F. Chen, Ta-Kai Chou, Yi-Wen Lin, Yu-Ten Ju, Shinn-Chih Wu, Ruoh-Fang Yan, Chyng-Yann Shiue, Sheau-Huei Chueh, Christer Halldin, Cheng-Yi Cheng, Kuo-Hsing Ma

**Affiliations:** 1Graduate Institute of Medical Sciences, National Defense Medical Center, Taipei 114, Taiwan; k6520319@yahoo.com.tw; 2Department of Nuclear Medicine, Tri-Service General Hospital, Taipei 114, Taiwan; treasure316@gmail.com (C.-H.C.); dakaichou@gmail.com (T.-K.C.); 3Graduate Institute of Life Sciences, National Defense Medical Center, Taipei 114, Taiwan; t70cyy@gmail.com; 4Institute of Oral Biology, National Yang-Ming University, Taipei 112, Taiwan; ywlin7@yahoo.com.tw; 5Department of Animal Science and Technology, National Taiwan University, Taipei 106, Taiwan; ytju@ntu.edu.tw (Y.-T.J.); scw01@ntu.edu.tw (S.-C.W.); 6Department of Nuclear Medicine, National Taiwan University Hospital, Taipei 100, Taiwan; rfyen@ntu.edu.tw (R.-F.Y.); shiue@ntuh.gov.tw (C.-Y.S.); 7Department of Biochemistry, National Defense Medical Center, Taipei 114, Taiwan; shch@ndmctsgh.edu.tw; 8Department of Clinical Neuroscience, Karolinska Institutet, 171 77 Stockholm, Sweden; christer.halldin@ki.se; 9Department of Biology and Anatomy, National Defense Medical Center, Taipei 114, Taiwan

**Keywords:** sertoli cell, ventral mesencephalic tissue, transplantation, positron emission tomography, Parkinson’s disease

## Abstract

Intra-striatal transplantation of fetal ventral mesencephalic (VM) tissue has a therapeutic effect on patients with Parkinson’s disease (PD). Sertoli cells (SCs) possess immune-modulatory properties that benefit transplantation. We hypothesized that co-graft of SCs with VM tissue can attenuate rejection. Hemi-parkinsonian rats were generated by injecting 6-hydroxydopamine into the right medial forebrain bundle of Sprague Dawley (SD) rats. The rats were then intrastriatally transplanted with VM tissue from rats or pigs (rVM or pVM), with/without a co-graft of SCs (rVM+SCs or pVM+SCs). Recovery of dopaminergic function and survival of the grafts were evaluated using the apomorphine-induced rotation test and small animal-positron emission tomography (PET) coupled with [^18^F] DOPA or [^18^F] FE-PE2I, respectively. Immunohistochemistry (IHC) examination was used to determine the survival of the grafted dopaminergic neurons in the striatum and to investigate immune-modulatory effects of SCs. The results showed that the rVM+SCs and pVM+SCs groups had significantly improved drug-induced rotational behavior compared with the VM alone groups. PET revealed a significant increase in specific uptake ratios (SURs) of [^18^F] DOPA and [^18^F] FE-PE2I in the grafted striatum of the rVM+SCs and pVM+SCs groups as compared to that of the rVM and pVM groups. SC and VM tissue co-graft led to better dopaminergic (DA) cell survival. The co-grafted groups exhibited lower populations of T-cells and activated microglia compared to the groups without SCs. Our results suggest that co-graft of SCs benefit both xeno- and allo-transplantation of VM tissue in a PD rat model. Use of SCs enhanced the survival of the grafted dopaminergic neurons and improved functional recovery. The enhancement may in part be attributable to the immune-modulatory properties of SCs. In addition, [^18^F]DOPA and [^18^F]FE-PE2I coupled with PET may provide a feasible method for in vivo evaluation of the functional integrity of the grafted DA cell in parkinsonian rats.

## 1. Introduction

Parkinson’s disease (PD) is typified by selective degeneration of dopaminergic (DA) neurons in substantia nigra pars compacta and is a prevalent neurodegenerative disorder [1]. Symptoms of PD include bradykinesia, resting tremor, postural instability, and ultimately rigidity [1]. Pharmacological and surgical treatment, such as deep brain stimulation, can partially alleviate these symptoms [2] but do not stop disease progression. Furthermore, the effectiveness of medications can decrease with time, and side effects such as paresthesia, depression, and dyskinesia may occur after long-term treatment [3,4].

Cell replacement therapy is a promising therapeutic strategy for PD and aims to re-establish DA function via re-introduction of exogenic DA cells into the affected brain circuit. Intra-striatal transplantation of cells harvested from human fetal ventral mesencephalic (hVM) tissue have been shown to restore striatal dopamine concentration in PD patients and provide long-lasting relief of the symptoms [5,6,7]. However, hVM tissues are unlikely to become a major PD therapy in that the supply of human fetal tissues is limited and there are serious ethical issues to consider. Alternatively, DA cells derived from human embryonic stem cells (hESCs) and induced pluripotent stem cells (iPSCs) can, in theory, resolve the supply issue and thus have been heavily tested in PD animal models [8,9,10]. Although hESCs and iPSCs have a strong potential for PD therapy, safety concerns with the use of these cells, including incomplete and unsynchronized differentiation, tumorigenesis, and neural overgrowth, are obstacles that have prevented their widespread use [11,12].

Fetal ventral mesencephalic (VM) tissue have been shown to have multiple benefits for PD cell therapy. Grafted DA cells from hVM tissue can re-innervate the striatum, and also release dopamine at high rate with auto-modulation via dopamine receptors [13,14,15]. Moreover, grafted hVM tissue can survive in patients for as long as 8–16 years [16,17]. Patients treated with this method had striatal dopamine concentration restored and also exhibited improvement of behavior. Despite all of the benefits of hVM tissue, its limitations have made the search for alternative cell sources, especially from non-human animals, an essential aim of PD cell therapy.

Porcine VM (pVM) tissue has become a promising cell source for PD therapy for a number of reasons. The supply of porcine embryos is abundant, and they are available whenever required. Pigs for transplantation can be raised in a specific pathogen–free environment [18,19]. DA cell grafts from pVM have shown no tumorigenesis in PD animal models and human patients [20,21]. Moreover, DA cells derived from pVM grafts can re-innervate the denervated striatum of patients, leading to recovery of motor function [22].

However, as in all xenotransplantation, immune suppression is required for pVM transplantation. Tacrolimus, neural, and cyclosporine are commonly used immunosuppressants for xenotransplantation, but side effects such as hepatotoxicity, nephrotoxicity, neurotoxicity, osteoporosis, diabetes, and hypertension are not uncommon [23,24,25]. Furthermore, studies have shown that immunosuppression drugs do not provide sufficient protection against immune rejection in the grafted area when treating PD [26,27,28]. To overcome this obstacle, we sought to identify a cell resource that could attenuate the immune response and provide benefits for DA cell survival.

Sertoli cells (SCs) are located in testis, which is an immune-privileged site that prevents germ cells from triggering an immune response [29]. SCs secrete multiple immune modulating molecules, such as the Fas ligand and transforming growth factor (TGF)-β [30]. Use of SCs in transplantation of other tissues and/or organ leads to a significant decrease in the levels of pro-inflammatory mediators, corresponding with increased anti-inflammatory cytokines at the grafted area [31,32]. Co-transplantation of SCs and tissues has resulted in prolonged survival of allograft and xenograft tissues [31,33,34]. Currently, SCs have been applied to a wide range of transplantation procedures, such as skin transplantation [31], cardiac grafts [34], and hepatocyte xenografts [33]. In addition, SCs hold great therapeutic potential for the treatment of neurodegeneration diseases. For example, rat ventral mesencephalic (rVM) tissue co-cultured with rat or porcine SCs results in enhanced neuronal survival and neuronal differentiation in vitro [35]. Study has also investigated the immune-modulatory effect in a hemiparkinsonian mouse model after co-grafting with rVM and rat SCs or mouse VM and rat SCs [36]. Another study showed that the neurite outgrowth was significantly increased in neurons co-cultured with SCs [37]. Taken together, these results suggest that SCs might be beneficial in cell replacement therapy for the treatment of PD.

The purpose of this study was to determine whether SCs are capable of enhancing the survival of VM allografts [18] and/or xenografts (porcine) in a PD model rat, and if so to reveal the underlying mechanisms of this enhancement. In order to determine DA neuron survival, we used small animal positron emission tomography (PET) with 2 [^18^F]-labeled radioligands, [^18^F]DOPA and [^18^F]FE-PE2I, to study dopamine synthesis and the dopamine transporter, respectively. [^18^F]FE-PE2I has been successfully used as an imaging agent to examine disease progression in PD patients [38]. We observed the immune response and DA cell survival of different transplantation groups via PET imaging and immunohistochemistry (IHC) studies. In addition, we used an apomorphine-induced rotation test to evaluate recovery of dopamine neuronal function.

## 2. Material and Methods

### 2.1. Animals

The Institutional Animal Care and Use Committee (IACUC) of the National Defense Medical Center (NDMC, Taipei, Taiwan) approved all of the animal experiment protocols (project identification code: IACUC-16-095 and IACUC-17-114; date of approval: August 1, 2016 and August 1, 2017). All animal experiments were conducted in accord with the Guidelines for the Use and Care of Laboratory Animals in Experimental Studies. Male Sprague–Dawley (SD) rats (eight weeks old, 280–300 g) were purchased from BioLASCO Ltd. (Taipei, Taiwan). Animals were housed in the NDMC animal facility under a 12 h light/dark cycle (light from 07:00 to 19:00), at a constant temperature of 23 ± 2 °C. The animals were given ad libitum access to a complete pellet diet and tap water. Lee–Sung pigs were obtained from National Taiwan University (Taipei, Taiwan), and raised in a pig house with an ISO9001-2015 certification by the British Standards Institution (certification number: FS686535). The experimental schedule is shown in Figure 1.

### 2.2. Hemiparkinsonian Rat Model

A hemiparkinsonian rat model was generated as previously described [39]. Briefly, eight-week-old male SD rats were deeply anesthetized with Zolitel (20 mg/kg, i.p.), and placed in a stereotactic apparatus. Then, 6-OHDA (20 μg in 4 μL of 0.02% ascorbic acid–holding saline) (Sigma-Aldrich, Saint Louis, MO, USA) was delivered unilaterally to the medial forebrain bundle. The coordinates were 7.8, 1.2, and 4.4 mm below the dura, lateral to the midline, and posterior to the bregma, respectively.

### 2.3. Behavioral Test

Severity of DA pathway damage in rats was evaluated by the apomorphine-induced rotational behavior test two weeks after the 6-OHDA lesion was created [40]. Rats with lesions were administered apomorphine subcutaneously at a dose of 0.5 mg/kg in 0.2% ascorbic acid–holding saline (Sigma-Aldrich, Saint Louis, MO, USA). Subsequently, drug-induced rotational responses were recorded by a rotometer system (MED Associates, Inc., St. Albans, VT, USA) [41]. A hemiparkinsonian disorder was defined as rats that rotated over four turns/min away from the lesion side. These rats were selected for transplantation experiments. Four weeks after transplantation, the rotation test was performed again to evaluate recovery of DA function.

### 2.4. SC Isolation

Rat SCs were isolated as previously described, with minor modifications [42]. Briefly, testes were surgically removed from 21-day-old (d21) SD rats. The seminiferous tubules were dispersed into 40 mL 1X HBSS, and washed two times with 1X HBSS followed by incubation with 25 mL of 0.05% collagenase solution in 1X HBSS (C2674, Sigma-Aldrich, Saint Louis, MO, USA) at 37 °C for 15 min, with shaking at 80 oscillations/min. The tubules were subsequently washed three times with 50 mL of 1X HBSS and thereafter incubated with 25 mL of 0.05% trypsin (T5266, Sigma-Aldrich, Saint Louis, MO, USA) in 1X HBSS at 37 °C for 10 min. Next, the tubules were washed with a 0.03% trypsin inhibitor (T6522, Sigma-Aldrich, Saint Louis, MO, USA) solution (20 mL 1X HBSS, 5–10 min) and then incubated in a multi-enzyme solution (37 °C, 40 min, shaking at 80 oscillations/min). The multi-enzyme solution contained 0.03% trypsin inhibitor, 25 mL of 1X HBSS, 0.1% collagenase, 0.04% DNase I (DN25, Sigma-Aldrich, Saint Louis, MO, USA), and 0.2% hyaluronidase (H6254, Sigma-Aldrich, Saint Louis, MO, USA). The derived supernatant was sieved via a 70 mm–pore size nylon mesh and centrifuged at 200× *g* for 10 min to derive a pellet of SCs. Finally, the pellet was washed three times with 1X HBSS and used for the experiments.

After SC isolation, IHC staining was used to confirm that the cells in pellet were indeed SCs, as numerous cells are stained with both a nuclear biomarker (nuclear red) and SC biomarker (follicle stimulating hormone receptor; FSHr) (Figure 2a–c). The cells were first stained with rabbit-anti FSHr (1:250; Aviva Systems Biology Corporation, San Diego, CA, USA), and then incubated with Alexa488-conjugated donkey anti-rabbit IgG (1:250; Jackson ImmunoResearch Laboratories, West Grove, PA, USA). Finally, the cells were stained with nuclear red (1:1000; AAT Bioquest, Inc., Sunnyvale, CA, USA). SCs were identified as being double-positive (FSHr^+^/nuclear red^+^). Flow cytometry was then used to isolate SCs from the cell pellet and to estimate the purity of SCs by calculating the percentage of FSHr positive cells (Figure 2d,e). The results indicated that approximately 80% of the cells isolated from the testis were SCs.

### 2.5. Mesencephalic Tissue Preparation and Transplantation

VM tissues used to establish allotransplantation and xenotransplantation models were obtained from embryonic day 14 SD rats and embryonic day 27 Lee-Sung pigs [39,43,44]. Dissection areas were selected according to a previous study, with some modifications [40,45]. The dissected tissues containing abundant DA cell bodies were kept in 1X HBSS. VM tissue was cut to small sections and subsequently grafted into the lesioned striatum using glass micropipettes, with the coordinates 2.5, 0.5, and 5.5 mm in length lateral to the midline, posterior to the bregma, and below the dura, respectively. Thirty-three hemiparkinsonian rats were divided into six groups, and different combinations of tissues were grafted into the striatum. (1) The sham group (n = 3) was injected with 4 μL 1X HBSS. (2) The SCs group (n = 6) received ~1.25 × 10^5^ SCs. (3) The rVM group (n = 6) was transplanted with rVM tissue. (4) The pVM group (n = 6) was transplanted with pVM tissue. (5) The rVM + SCs group (n = 6) was co-grafted rVM tissue and SCs (~1.25 × 10^5^ cells). (6) The pVM + SCs group (n = 6) was co-grafted pVM tissue and SCs (~1.25 × 10^5^ cells).

### 2.6. Radiopharmaceuticals

[^18^F] DOPA was synthesized and provided by the Department of Nuclear Medicine affiliated with National Taiwan University Hospital. [^18^F] FE-PE2I was synthesized as previously reported, with some modifications [46]. Briefly, nucleophilic fluorination of a tosyl precursor was performed in dimethyl sulfoxide with dried K [1 8F]/K_2.2.2_, followed by modified HPLC purification (without a pre-purified cartridge). The desired compound was obtained after solid phase extraction and formulation in phosphate buffered saline.

The non-decay corrected radiochemical yield for [^18^F] FE-PE2I was 4.98% ± 1.73% (n = 15), with a radiochemical purity > 96%. The specific activity was 179 ± 52 GBq/μmol at the time of end-of-synthesis (EOS). The injected mass of [^18^F] FE-PE2I was about 94.2 ng/kg in subsequent imaging studies, with intravenous injection of 18–22 MBq of [^18^F] FE-PE2I.

### 2.7. Small Animal PET Imaging

The protocol for small animal PET imaging was adapted from previous research [47]. PET imaging was performed with a small animal PET scanner (BIOPET 105, BIOSCAN, Santa Clara, CA, USA) at one week before the 6-OHDA lesion was created, two weeks after the 6-OHDA lesion was created, and four weeks after transplantation. [^18^F] DOPA (22.2–25.9 MBq; 0.6–0.7 mCi) and [^18^F] FE-PE2I (14.8–18.5 MBq; 0.4–0.5 mCi) was administrated via the tail vein of the rats. Thirty minutes before [^18^F] DOPA injection, entacapone (Toronto Research Chemicals, Toronto, ON, Canada; 10 mg/kg) and carbidopa (Toronto Research Chemicals, Toronto, ON, Canada; 10 mg/kg) were administrated intraperitoneally [48,49]. The SD rats were anesthetized by passive inhalation of an isoflurane–oxygen mixture (2% and 5% isoflurane for maintenance and induction, respectively). PET image collection was executed 50 min ([^18^F] DOPA) and 20 min ([^18^F] FE-PE2I) after radioligand injection. The collection process was performed at an energy window of 250–700 keV. Image reconstruction was performed using a 2D filtered back-projection (ramp filter, with the cutoff being determined at Nyquist frequency) and the Fourier re-binning algorithm. Amide software (stanford university, Santa Clara, CA, USA) was used for analysis of the PET images. The rat brain atlas and magnetic resonance imaging (MRI) results were used for brain region confirmation [50]. The atlas and MRI results were also used to delineate volumes of interest of the cerebellum and striatum as observed in the PET images that had been reconstructed and summated. The specific uptake ratio (SUR) was expressed as (striatum − cerebellum)/cerebellum.

### 2.8. IHC Staining

Four weeks after transplantation, rats from each group were sacrificed for IHC experiments [43]. Rat brains were removed and post-fixed with 4% paraformaldehyde overnight at 4 °C. Brains were then soaked in 20% sucrose in 0.1M PBS for two days and then 30% sucrose in 0.1M PBS for two days. Brains were sliced into a series of coronal sections (30 µm) on a Cryostat Microtome (Leica CM 3050; Leica Microsystem, Wetzlar, Germany). All brain slices (approximately 90 slices per animal) that contained grafted region were divided into four sets, and prepared for tyrosine hydroxylase (TH), dopamine transporter (DAT) [51], CD3, and major histocompatibility complex (MHC) class II (OX-6)/ionized calcium binding adaptor molecule 1 (Iba1) staining. The survival of DA neurons in grafted regions was evaluated by immunostaining with anti-TH and anti-DAT antibodies. Anti-CD3 antibody was used to identify infiltrating T-cells in the grafted region. Anti-Iba1 and anti-OX-6 antibodies were used to calculate the total and activated microglia in the grafted striatum, respectively. 3,3-diaminobenzidine (DAB) was used in IHC staining of TH, DAT, and CD3; a positive reaction was identified as a brown color in brain slices.

First, brain slices were rinsed with blocking solution containing 3% normal goat serum (Vector, Burlingame, CA, USA) in PBS, and 0.5% Triton X-100 (Sigma-Aldrich, Saint Louis, MO, USA). The rinsed slices were incubated overnight at 4 °C with rabbit anti-CD3 antibody (1:200; Abcam, Cambridge, UK), rabbit anti-TH antibody (1:2000 dilution; Millipore Corporation, Billerica, MA, USA), or rabbit anti-DAT antibody (1:500; Abcam, Cambridge, UK). They were then incubated for 1 h with goat anti-rabbit biotinylated IgG (1:200; Vector). After washing three times, the slices were incubated for 1 h with avidin–biotin complex at a 1:200 dilutions (Vectastain ABC kit, Vector, Burlingame, CA, USA). For visualization, 0.05% DAB was applied for 6 min, and then the slices were mounted on glass slides.

The VM tissue distribution in the transplantation side striatum served as the basis for the definition of the grafted area. The optical density (OD) of the grafted areas was measured on all IHC sections. Images of the striatum stained with different antibodies were acquired, and examined using a slide scanner (Axio Scan.Z1; ZEISS, Oberkochen, Germany) and ImageJ software (version 1.8.0, NIH, Bethesda, MD, USA, 2014). The OD ratio was calculated as follows: OD ratio = (OD_striatum_−OD_corpus callosum_)/OD_corpus callosum_. The numbers of TH immunoreactive (TH-ir), DAT-ir, and CD3-ir cells of grafted regions were calculated and then divided by the areas, and they were further divided into 30 µm. Therefore, data of cell number was expressed as the number of cells per cubic millimeter.

Microglia activation was determined using immunofluorescence staining. Brain slices were double-stained with rabbit anti-Iba1 (1:500; Wako, Osaka, Japan) and mouse anti-OX-6 (1:150; Abcam, Cambridge, UK) and then stained with Alexa488-conjugated donkey anti-rabbit IgG (1:250; Jackson ImmunoResearch Laboratories, West Grove, PA, USA) and cy3-conjugated donkey anti-mouse IgG (1:250; Jackson ImmunoResearch Laboratories, West Grove, PA, USA). Finally, the brain sections were stained with nuclear red (1:1000; AAT Bioquest, Inc., Sunnyvale, CA, USA) for nuclear quantitation. The fluorescence image was captured by confocal microscopy (LSM880; Zeiss, Oberkochen, Germany). The quantification of microglia or SCs was performed as previously described [51]. Six consecutive brain sections containing graft areas were chosen, and the number of microglia in the whole graft was calculated for each brain section. The size of the grafted areas was 850.2 µm^2^. The total number of microglia and activated microglia was divided by 850.2 µm^2^ and expressed as the number per square millimeter.

### 2.9. Statistical Analysis

Data were presented as mean ± standard deviation. PET image SURs, behavioral test results, OD ratios and cell counts in the grafted areas derived through IHC staining were compared between groups. ANOVA with Bonferroni post-test was used for multiple comparisons, and Student’s *t*-test was used to compare the results of two independent groups. Values of *p* < 0.05 were considered to indicate statistical significance.

## 3. Results

### 3.1. SCs Enhanced the Effect of VM Allotransplantation on DA Functional Recovery in Hemiparkinsonian Rats

The apomorphine-induced rotation test was used to evaluate DA function in hemiparkinsonian rats before and after transplantation (Figure 3). An increase in net rotation was noted in hemiparkinsonian rats, whereas no behavioral improvement was observed in the sham (n = 3) and SCs groups (n = 6). The hemiparkinsonian rats that received VM tissue from both rats (allotransplantation, rVM) (n = 6) and pigs (xenotransplantation, pVM) (n = 6) exhibited significant DA recovery. Co-transplantation of SCs in allo- and xenotransplantation of VM tissue led to improved DA function in the rats (n = 6/group). Moreover, rats that received rVM+SCs exhibited less rotations compared to rVM-grafted rats, indicating SCs as a co-graft material could enhance the effect of VM allotransplantation. However, the same effect was not observed with SCs in xenotransplantation.

### 3.2. SCs Enhanced the Functional Result of DA Allografts and Xenografts in the Striatum of Hemiparkinsonian Rats as Determined by [^18^F]DOPA Coupled with Small Animal PET

The functional results of DA allo- and xeno-transplantation in hemiparkinsonian rats were evaluated by [^18^F] DOPA coupled with small animal PET. The PET images were acquired before and after creation of the 6-OHDA lesion and after transplantation (Figure 4a). A 6-OHDA lesion was created in the right median forebrain bundle of male SD rats to establish hemiparkinsonian rats, whose striatal [^18^F] DOPA uptake level on the lesioned side (left side of the brain sections, Figure 4a) was decreased (middle column of each group, Figure 4a). Weak recovery of uptake was found in the rVM (n = 6), pVM (n = 6), SCs+rVM (n = 6), and SCs+pVM groups (n = 6) (right column of each group, Figure 4a). To measure the effects of transplantation and 6-OHDA lesion, the SUR of [^18^F] DOPA on the lesioned side of striatum was used for further analysis (Figure 4b). After 6-OHDA injection, the SUR of [^18^F] DOPA of the lesioned striatum dropped 85–95% from the baseline value. In sham (n = 3) and SC group (n = 6), there was no difference is the SUR of [^18^F] DOPA, suggesting SCs alone do not induce recovery of DA cells. However, the uptake of [^18^F] DOPA significantly increased in the rat (allograft) and porcine (xenograft) VM tissue transplantation groups. SCs co-grafted with rat or porcine VM tissue led greater uptake of [^18^F] DOPA than when SCs were not co-grafted (both, *p* < 0.05). These results suggest that SCs can enhance the outcome intra-striatal allo- and xeno-transplantation of VM tissue grafts in hemiparkinsonian rats.

### 3.3. SCs Enhance Maturation of DA Allografts and Xenografts in the Striatum of Hemiparkinsonian Rats as Determined by [^18^F]FE-PE2I Coupled with Small Animal PET

[^18^F] FE-PE2I coupled with small animal PET is a good method for the in vivo measurement of dopamine transporter expression and was used to evaluate the maturation of the DA neural grafts in the rat striatum [52]. Similar to the [^18^F]DOPA imaging results, striatal [^18^F] FE-PE2I uptake levels on the lesioned side were decreased after the lesion was created (middle column of each group, Figure 5a), and a slight recovery of the uptake was observed in the rVM (n = 6), pVM (n = 6), SCs+rVM (n = 6), and SCs+pVM groups (n = 6) after transplantation (right column of each group, Figure 5a). Results of the quantitative analysis of the SURs are shown in Figure 5b. The SURs of [^18^F] FE-PE2I were decreased after intracerebral injection of 6-OHDA in all groups. Transplantation of HBSS (n = 3) or SCs (n = 6) in the lesioned striatum did not induce a significant change in [^18^F] FE-PE2I uptake. This result is consistent with the [^18^F] DOPA imaging results, i.e., SCs are unable to induce recovery of DA cells. In contrast, there was no significant difference in the SURs of [^18^F] FE-PE2I in the rVM group and pVM group after transplantation. In addition, co-grafts of SCs with VM tissue in allo- and xeno-transplantation groups resulted in significant uptake of [^18^F] FE-PE2I in the grafted striatum (Figure 5a). This result suggests that SCs co-grafted with VM tissue can enhance maturation of transplanted DA cells in allo- and xeno-transplantation.

### 3.4. SCs Enhance the Survival of DA Cells in Allo- and Xeno-Transplantation

To determine whether the higher SURs of [^18^F] DOPA and [^18^F] FE-PE2I were due to higher DA cell survival in the grafted striatum, we performed TH staining of the grafted striatum. In photomicrographs of brain sections of the sham (n = 3) and SCs (n = 6) groups, TH-ir (DA) cells or fibers in the grafted striatum (right side of the brain sections, Figure 6a,b) were hardly visible. In contrast, DA cell bodies and fibers were visible in the grafted striatum of the rVM (n = 6) (Figure 6c), rVM+SCs (Figure 6d) (n = 6), pVM (n = 6) (Figure 6e), and pVM+SCs (n = 6) (Figure 6f) transplantation groups.

We then quantified these IHC results with two different methods. First, we counted the number of TH-ir cell bodies in the grafted striatum (Figure 6g). The DA cell body densities of the rVM, pVM, rVM+SCs, and pVM+SCs groups were 1341 ± 269/mm^3^, 1050 ± 187/mm^3^, 2130 ± 385/mm^3^, and 1598 ± 398/mm^3^, respectively. Using the sham group (22 ± 2/mm^3^) as the standard, we found that all four groups contained abundant DA cell bodies over the grafted striatum after transplantation. Moreover, the co-grafted groups (rVM+SCs and pVM+SCs) had higher DA cell body densities in the grafted striatum than the single grafted groups (rVM or pVM). Second, we used OD ratio to evaluate the density of the DA cells (cell bodies + fibers) in the grafted striatum (Figure 6h). Analysis of the OD ratios was consistent with the cell body density results; the rVM+SCs and pVM+SCs groups had significantly higher DA cell density than the rVM and pVM group. These results suggested that SCs co-grafted with VM tissue enhanced the survival and neuritis growth of DA cell grafts. Furthermore, we found that grafted SCs coexisted with DA cells in the brains of the rVM+SCs and pVM+SCs groups at the fourth week after the transplantation (Figure 7). This observation indicated that SCs themselves can survive in the host brain and might be able to provide long-term beneficial effects to the grafts.

### 3.5. SCs Enhance DA Cell Maturation in Both Allo- and Xeno-Transplantation

The results of the [^18^F] FE-PE2I small animal PET experiment suggested that SCs could enhance the maturation of grafted DA cells in the striatum. To confirm this finding, we performed DAT immunostaining of the brain sections. Consistent with the PET results, very few DAT-ir cells were found in the grafted striatum of the sham (n = 3) and SCs groups (n = 6) (Figure 8a,b,g). DAT-ir cells were observed in the grafted striatum of the rVM (n = 6), rVM+SCs (n = 6), pVM (n = 6), and pVM+SCs groups (n = 6) (Figure 8c–f). Quantification of the IHC results showed that the sham group only contained 12 ± 2/mm^3^ DAT-ir cells in the grafted striatum (Figure 8g). The rVM and pVM groups contained 331 ± 81/mm^3^ and 279 ± 44/mm^3^ DAT-ir cells, respectively. The rVM+SCs and pVM+SCs groups contained 1392 ± 145/mm^3^ and 1122 ± 144/mm^3^ DAT-ir cells, respectively. The DAT-ir cell body density in grafted striatum of the four groups was significantly higher than that of the sham group (all, *p* < 0.05). The co-grafts were consistently better than single grafts in generating mature DA cells in the grafted striatum. This result may explain the results of the [^18^F] FE-PE2I small animal PET imaging (Figure 5). Furthermore, it is possible that SCs, when co-grafted with VM tissues (allotransplantation or xenotransplantation), can produce beneficial factors that enhance maturation of transplanted DA cells [35].

### 3.6. SCs Attenuate the Immune Response of Microglia in Grafted Striatum after Allo- and Xeno-Transplantation

The enhancement effect of SCs on DA cell survival is likely due to immune-modulatory properties of SCs. We evaluated the immune response in grafted striatum via immunostaining brain slices with Iba1 (total microglia) and OX6 (activated microglia). The microglia cell numbers were higher after transplantation in all groups, except the sham group (n = 3) and the SCs group (n = 6) (Figure 9c–f). For quantification, we calculated the density of total microglia and activated microglia per mm^2^ in the grafted striatum and contralateral striatum (Figure 9g). In the contralateral striatum, the density of total microglia and activated microglia were approximately 150/mm^2^ and 5/mm^2^, respectively. In hemiparkinsonian rats that received rVM (n = 6) or pVM tissue (n = 6), the total microglia in the grafted striatum were 967 ± 467/mm^2^ and 1171 ± 650/mm^2^, and the activated microglia were 630 ± 277/mm^2^ and 772 ± 379/mm^2^, respectively. In rats that received rVM+SCs (n = 6) or pVM+SCs (n = 6), the total microglia in the grafted striatum were 339 ± 71/mm^2^ and 456 ± 88/mm^2^, and the activated microglia were 107 ± 21/mm^2^ and 219 ± 73/mm^2^, respectively. Moreover, rats that received rVM and pVM co-grafted with SCs had a lower OX6+/Iba1+ ratio than rats in the rVM group and pVM group (Figure 9h). These results indicated that co-grafting with SCs can provide immune-modulatory effects via attenuating activated microglia in allo- and xeno-transplantation.

### 3.7. SCs Attenuate the T-Cell Infiltration Immune Response of after Allo- and Xeno-Transplantation

To investigate whether the immune-modulatory effect of SCs was via attenuating T-cell infiltration in the grafted striatum, we calculated the density of T-cells using the CD3 biomarker. The CD3-ir cell number was counted in the grafted striatum (Figure 10a–f) and the contralateral striatum (Figure 10g–l). The results indicated the T-cell number was higher in the rVM (n = 6), pVM (n = 6), and pVM+SCs groups (n = 6). For quantification, the T-cell densities of the contralateral side were used as the standard. The T-cell density in contralateral striatum was approximately 200/mm^3^ in all of the experimental groups, and was similar to that of the sham group (207 ± 26/mm^3^). The T-cell densities were 809 ± 133/mm^3^, 322 ± 39/mm^3^, 1707 ± 445/mm^3^, and 446 ± 38/mm^3^ in the rVM, rVM+SCs (n = 6), pVM, and pVM+SCs group, respectively. The density was markedly increased in the allotransplantation (rVM) group and the xenotransplantation (pVM) group, compared with the density of the contralateral striatum. These results showed that SCs co-grafted with rVM and pVM tissues significantly attenuated the T-cell number in grafted striatum compared with the rVM and pVM groups. This suggested that SCs co-grafted with rVM or pVM tissues provide an immune-modulatory effect that markedly reduced T-cell infiltration in the grafted striatum.

### 3.8. The SURs of [^18^F]DOPA and [^18^F]FE-PE2I Were Highly Correlated

Pearson’s correlation was used to investigate the relations between the SURs of [^18^F]DOPA and [^18^F]FE-PE2I in the grafted striatum of the rVM (n = 3), rVM+SCs (n = 4), pVM (n = 4), and pVM+SCs (n = 4) groups (Figure 11). The SURs of each group at all time points were included in the analysis. The Pearson’s R square results were rVM group, R square = 0.9679; rVM+SCs group, R square = 0.9449; pVM group, R square = 0.9626; pVM+SCs group, R square = 0.9280. These results indicated the SURs of [^18^F] DOPA and [^18^F] FE-PE2I were highly correlated in all of the animal groups tested in this study.

## 4. Discussion

The present study showed that transplanting SCs with rat (allograft) or porcine (xenograft) VM tissues resulted in better functional recovery of drug-induced rotational behavior and better striatal uptake recovery of [^18^F] FE-PE2I and [^18^F] DOPA in hemiparkinsonian rats. In addition, a co-graft SCs and VM tissues (rat or porcine) led to an increase in the numbers of TH-ir and DAT-ir neurons in the grafted striatum. These results suggest that SCs enhance the survival of grafted dopaminergic neurons in vivo.

SCs have been shown to secret numerous trophic factors, including insulin-like growth factor-I (IGF-I), basic fibroblast growth factor (bFGF), and glial cell line-derived neurotrophic factor (GDNF) [53]. The beneficial effect of SCs observed in this study may be in part due to the release of GDNF, which enhances TH-positive cell survival and nerve fiber formation after transplantation of VM tissues in the striatum [54]. Previous study showed that grafted dopaminergic precursor cells may not differentiate completely after transplantation [54] and may require several neurotrophic factors, such as GDNF and bFGF, to differentiate into mature cells [55]. As mentioned above, SCs may be beneficial for the differentiation of dopaminergic neurons due to the release of GDNF and bFGF.

The transplantation-induced significant recovery in the striatal uptake of [^18^F] FE-PE2I was only found in the rVM+SCs and pVM+SCs groups, whereas significant recovery in the uptake of [^18^F] DOPA was found in the rVM+SCs, pVM+SCs, rVM alone, and pVM alone groups. This discrepancy may be because the 2 radioligands target different biomarkers. [^18^F] DOPA is a dopamine analogue, whereas [^18^F] FE-PE2I is a DAT imaging agent [56,57,58]. In the rat brain, DAT is regarded as a biomarkers of DA neuron maturation, whereas dopamine synthesis may be present as early as embryonic day 12.5 [59].

The numbers of OX6-positive and CD3-positve cells were increased in lesioned striatum of the hemiparkinsonian rats. Grafting rat or porcine VM tissue with SCs led to a significant decrease in the number of OX6- and CD3-positive cells, whereas this phenomenon was absent in VM alone grafted group. These findings suggested that SCs exert immune-modulatory effects in the grafted striatum of hemiparkinsonian rats. SCs are known to secrete several immune-stimulators that generate an immune-privileged microenvironment [29]. These cells modulate the T-cell response through transforming growth factor B (TGFB) and consequently influence Th1 and Th2 responses [29]. SCs have been reported to modulate the expression of pro-inflammatory cytokines, including IL-1 and IL-6, and to inhibit macrophage migration via Activin A [60].

Mesenchymal Stem cells (MSCs) is another promising cellular material for PD therapy. Similar to SCs, MSCs could secret neurotrophic factors, such as nerve growth factor (NGF), brain-derived neurotrophic factor (BDNF), and GDNF [61,62]. These factors could slow down degenerative progression of dopaminergic neurons and induce proliferation of neural stem cells [63]. MSCs have been grafted bilaterally into striatum of early stage PD patients, leading to improvement in Unified Parkinson’s Disease Rating Scale (UPDRS) scores of the patients. [64]. These neuroprotective effects of MSCs may result from their anti-inflammatory properties. The effects of MSCs on lipopolysaccharides (LPS)-induced microglial activation have been evaluated in an in vitro study, indicating that MSCs could inhibit activation of microglia, reduce production of TNF-α and inducible nitric oxide synthase (iNOS), and increase production of anti-inflammatory cytokine IL-10 and transforming growth factor β (TGF-β) [65]. These results suggest that MSCs might be used as a co-graft with the VM tissue in the cell therapy for late-stage PD patients.

## 5. Conclusions

The use of SCs as a co-graft material improves survival rates of VM allografts and xenografts in a parkinsonian rat model. This method circumvents the toxicity of immunosuppressants that may be used for inflammation-associated rejection and thus represents an advantageous approach. We also confirmed that [^18^F] DOPA and [^18^F] FE-PE2I coupled with PET is capable of monitoring the functional integrity of grafted DA cells. The co-graft of SCs and non-human VM tissues might provide a new therapeutic approach for the treatment of human PD.

## Figures and Tables

**Figure 1 cells-08-01420-f001:**
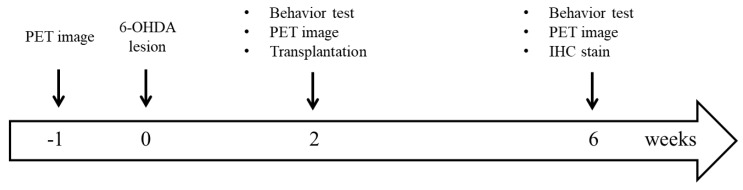
Experimental flowchart. [^18^F]-DOPA and [^18^F] FE-PE2I PET scans were performed at three time points (once before and twice after a unilateral 6-hydroxydopamine (6-OHDA) lesion was made to medial forebrain bundle of the animals). Behavior test refers to the apomorphine-induced rotation test. Transplantation was performed two weeks after the 6-OHDA lesion was made. At six weeks after the lesion, the rats were sacrificed for immunohistochemistry (IHC) studies.

**Figure 2 cells-08-01420-f002:**
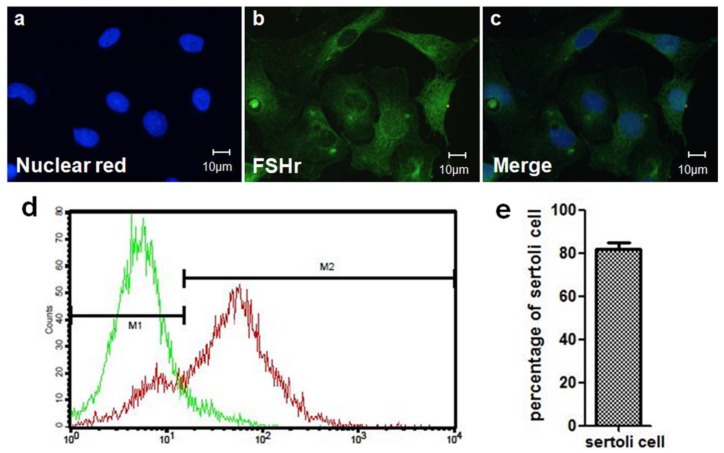
Isolation of Sertoli cells (SCs). IHC staining was used to identify SCs isolated from the testis. Staining included (**a**) nuclear red staining (biomarker of nucleus) and (**b**) immunostaining for FSHr (biomarker of SCs). (**c**) SCs were identified as double-labeled cells. (**d**) Flow cytometry showed different fluorescence intensity in M1 (cell suspension only stained with florescent secondary antibody) and M2 (cell suspension stained with FSHr primary antibody and florescent secondary antibody). The SCs (M2) exhibited a tremendously shifted peak as compared to the control (M1). (**e**) The purity of the SCs was calculated by flow cytometry.

**Figure 3 cells-08-01420-f003:**
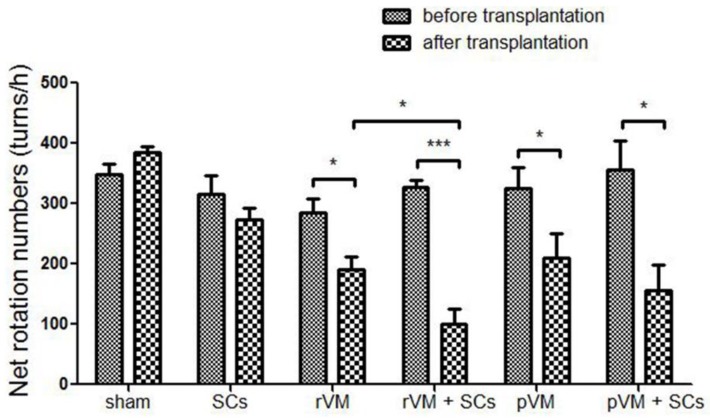
The apomorphine-induced rotational behavior test was used to evaluate dopaminergic (DA) function of hemiparkinsonian rats that received different neural grafts. The test was performed before and after transplantation. * *p* < 0.05; *** *p* < 0.001.

**Figure 4 cells-08-01420-f004:**
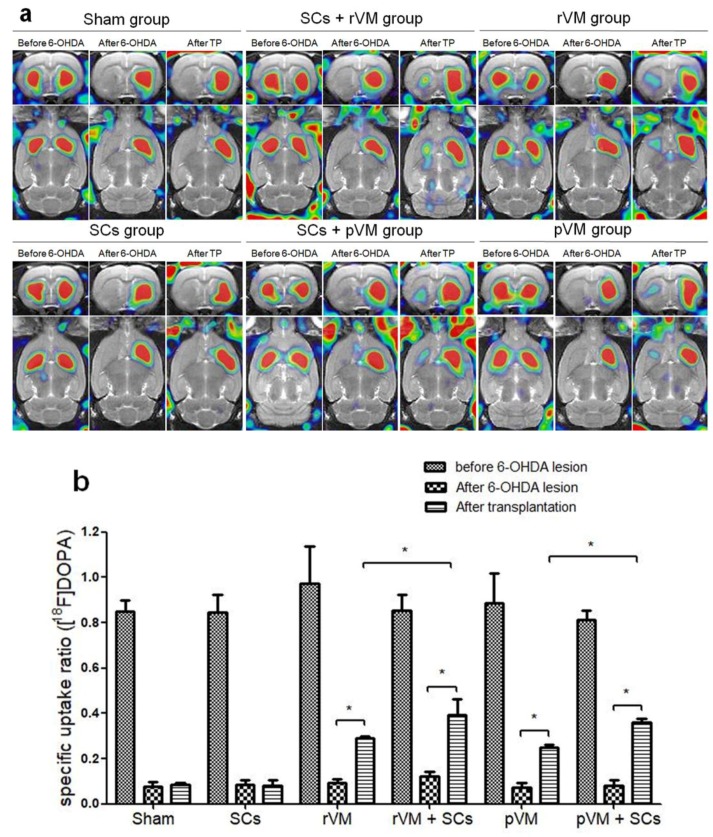
PET images of ^18^F-DOPA uptake distribution in rat brains. (**a**) Coronal (upper panel) and horizontal (lower panel) sections of each group were acquired 50–80 min after injection of ^18^F-DOPA. Left, middle, and right columns of each group represent the ^18^F-DOPA uptake before the 6-OHDA lesion, after the 6-OHDA lesion, and after transplantation with different grafting materials, respectively. (**b**) Specific uptake ratios (SURs) of ^18^F-DOPA of the grafted striatum at different time points and under different grafting conditions (* *p* < 0.05).

**Figure 5 cells-08-01420-f005:**
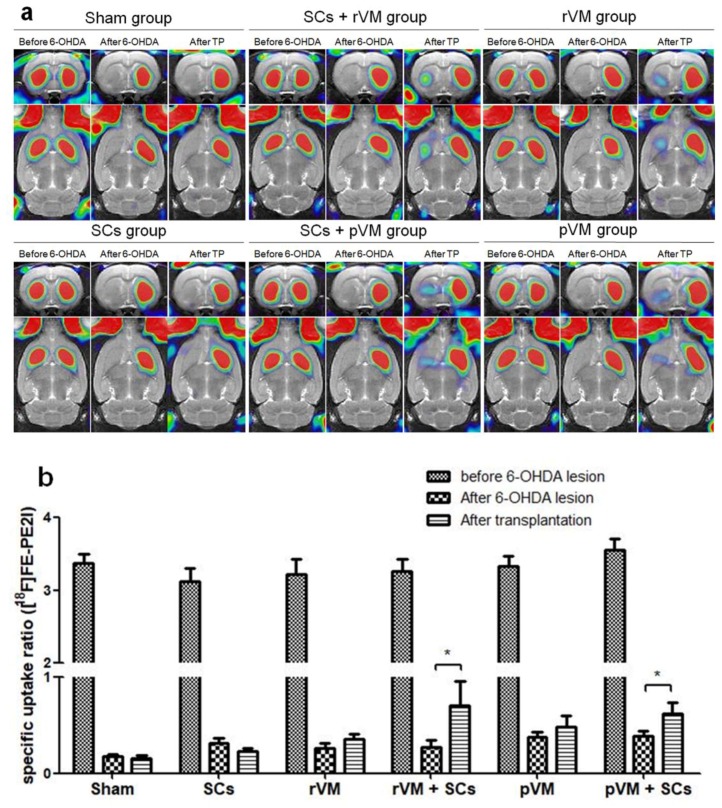
PET images of [^18^F] FE-PE2I uptake distribution in rat brains. (**a**) Coronal (upper panel) and horizontal (lower panel) sections of each group were acquired 20–40 min after injection of [^18^F]FE-PE2I. Left, middle, and right columns of each group represent [^18^F]FE-PE2I uptake before the 6-OHDA lesion, after the 6-OHDA lesion, and after transplantation with different grafting materials, respectively. (**b**) SURs of [^18^F] FE-PE2I in the grafted striatum at different time points and under different grafting conditions (* *p* < 0.05).

**Figure 6 cells-08-01420-f006:**
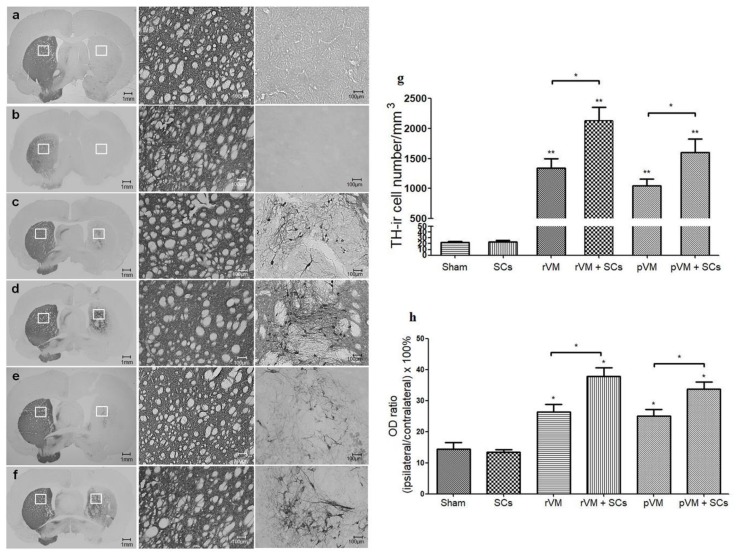
Photomicrographs of tyrosine hydroxylase immunoreactive (TH-ir) cells, and quantification results in hemiparkinsonian rat striatum at four weeks following transplantation. TH-ir cell bodies and fibers in the grafted side (right side of the coronal brain sections) of the rat ventral mesencephalic (rVM), rVM+SCs, pig ventral mesencephalic (pVM), and pVM+SCs groups. Greater densities of TH-ir cell bodies and fibers were noted in the rVM+SCs and pVM+SCs groups as compared to the rVM and pVM groups. Quantification data were consistent with the IHC results. (**a**) Sham group. (**b**) SC group. (**c**) rVM group. (**d**) rVM+SCs group. (**e**) pVM group. (**f**) pVM+SCs group. (**g**) Quantification of TH-ir cell density. (**h**) Optical density (OD) ratio in grafted striatum (* *p* < 0.05; ** *p* < 0.01).

**Figure 7 cells-08-01420-f007:**
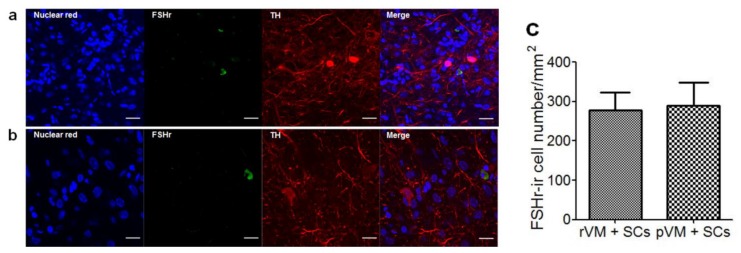
SCs and DA cells co-existed in the grafted striatum of the rVM+SCs and pVM+SCs groups four weeks after transplantation. Brain sections were stained with a dopaminergic neuron marker (TH) and SCs marker (FSHr). (**a**) rVM+SCs group. (**b**) pVM+SCs group. (**c**) Quantification data of FSHr-ir cell numbers. Scale bar = 20 μm.

**Figure 8 cells-08-01420-f008:**
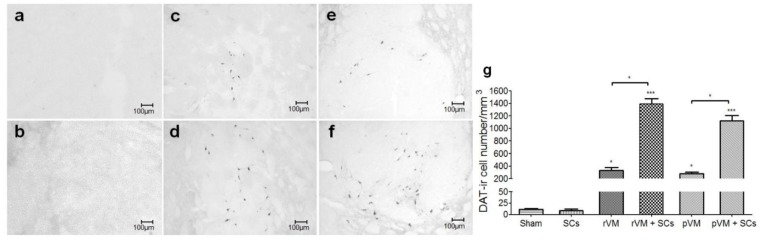
Photomicrographs of DAT-ir and quantification results at four weeks after transplantation. DAT-ir cell bodies in the grafted sides of the rVM, rVM+SCs, pVM, and pVM+SCs groups. The densities of DAT-ir cell bodies in the rVM+SCs and pVM+SCs groups were higher than those of the rVM and pVM groups. Quantification data were consistent with the IHC results. (**a**) Sham group. (**b**) SC group. (**c**) rVM group. (**d**) rVM+SCs group. (**e**) pVM group. (**f**) pVM+SCs group. (**g**) Quantification of DAT-ir cell density. (**h**) OD ratio in grafted striatum (* *p* < 0.05; *** *p* < 0.01). Scale bar = 100 μm.

**Figure 9 cells-08-01420-f009:**
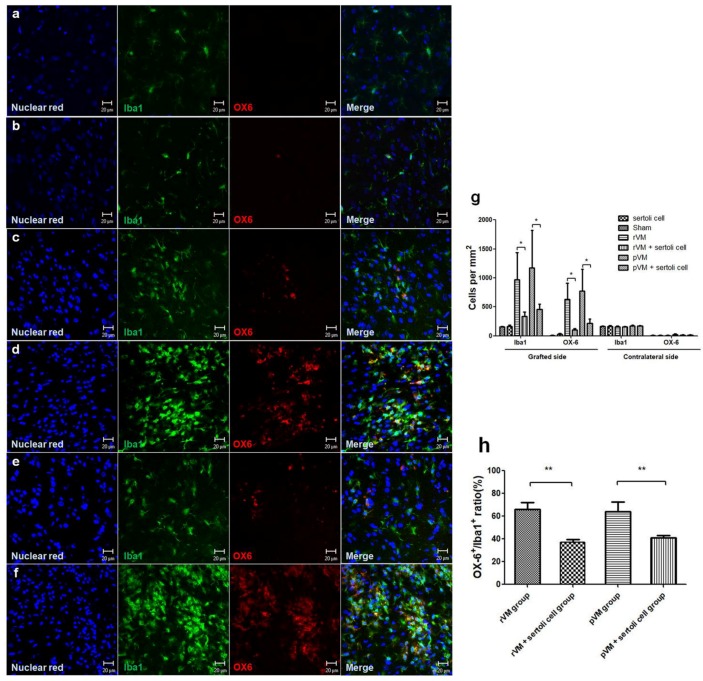
Immunofluorescence staining with anti-Iba1 and anti-OX6 antibody and quantification results at four weeks after transplantation. A severe immune response in grafted striatum was noted in the rVM group and pVM group. Co-grafted groups showed lower microglia cell numbers compared with the rVM and pVM groups. Quantification results showed that Iba1-ir and OX6-ir were obviously increased in the rVM and pVM groups. The OX-6+/Iba1+ ratio was lower in the rVM+SCs and pVM+SCs groups than in the rVM and pVM groups. (**a**) Sham group. (**b**) SCs group. (**c**) rVM group. (**d**) rVM+SCs group. (**e**) pVM group. (**f**) pVM+SCs group. (**g**) Quantification of Iba1-ir and OX-6-ir cell numbers. (**h**) OX-6+/Iba1+ ratio. Scale bar = 20 μm.

**Figure 10 cells-08-01420-f010:**
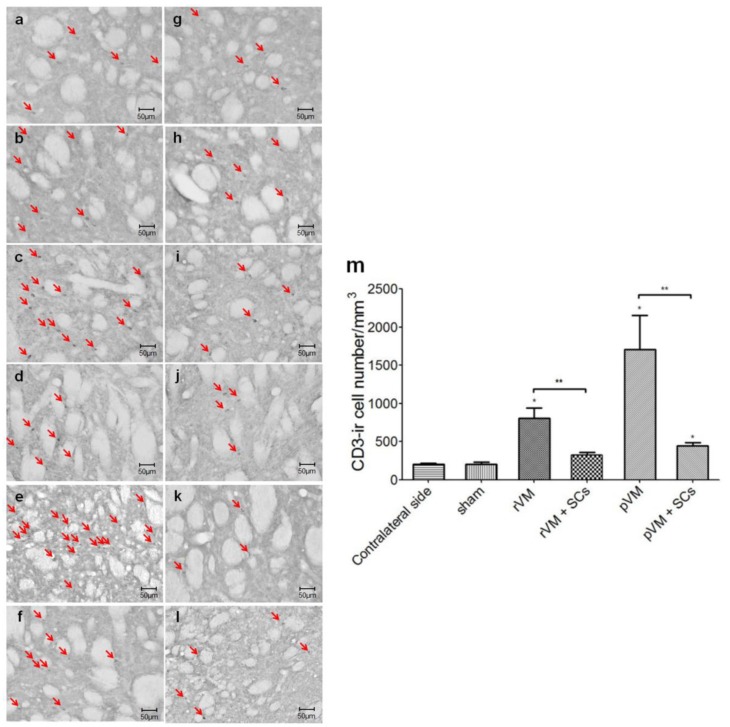
Grafted striatum and contralateral striatum stained with anti-CD3 antibody. Photomicrographs of grafted striatum: (**a**) Sham group, (**b**) SCs group, (**c**) rVM group, (**d**) rVM+SCs group, (**e**) pVM group and (**f**) pVM+SCs group. Photomicrographs of contralateral striatum: (**g**) Sham group, (**h**) SCs group, (**i**) rVM group, (**j**) rVM+SCs group, (**k**) pVM group and (**l**) pVM+SCs group. The number of CD3-ir cells in the rVM group and pVM group were greater than in the rVM+SCs group and pVM+SCs. (**m**) Quantification of CD3-positive cells in striatal grafts at four weeks after transplantation. The number of CD3-ir cells were increased in the grafted striatum of the rVM, pVM, and pVM+SCs groups at four weeks after transplantation (** *p* < 0.01, the grafted side compared to each contralateral side). The number of CD3-ir cell in the grafted striatum of the rVM+SCs and pVM+SCs groups was markedly less than in the rVM group and pVM group (* *p* < 0.05; ** *p* < 0.01). Scale bar = 50 μm.

**Figure 11 cells-08-01420-f011:**
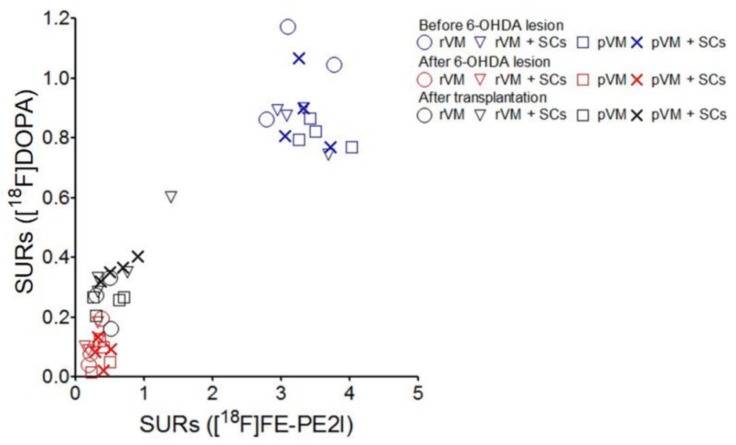
Pearson’s correlation analysis of ^18^F-DOPA and [^18^F]FE-PE2I in the grafted striatum of the rVM, rVM+SCs, pVM, and pVM+SCs groups. In all groups, Pearson’s R square was greater than 0.9, indicating a high correlation between the SUR of ^18^F-DOPA and [^18^F]FE-PE2I.

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
