# Peer review of "The Effect of Sertoli Cells on Xenotransplantation and Allotransplantation of Ventral Mesencephalic Tissue in a Rat Model of Parkinson’s Disease"

_cells, 2019, doi:10.3390/cells8111420_

Round 1
Reviewer 1 Report
In this manuscript Chen et al. investigated the effect of Sertoli cells on survival of the grafted dopaminergic neurons. They demonstrated that co-grafting of Sertoli cells and ventral mesencephalic tissue provided long-term beneficial effect to the grafts and improved the functional recovery in a rat model of Parkinson’s disease. They argued that these effects depended on immune-privileged microenvironment generated by Sertoli cells. This approach allows to avoid immunosuppressants and prevent the rejection.
This work is scientifically and well-written and presents interesting results although they lost originality. Different papers suggested the Sertoli transplantation as alternative treatment of neurodegenerative disorders such as Parkinson’s disease.
The research regarding the co-transplantation of dopaminergic neurons and Sertoli cells for the alternative treatment of neurodegeneration has been extensive.
Curiously, in a previous study by Chen et al, the allograft effect of Sertoli cells and rVM tissues in a PD rat model using [18F]FE-PE2I/ animal PET has been already evaluated. (J Nucl Med May 1, 2017 vol. 58 no. supplement 1 271).
They already showed the effect of cotransplantation of rVM and Sertoli on tyrosine hydroxylase, dopamine transporter, T-cell and microglia markers, as in the present study.
The release of neurotrophic factors such as GDNF/BDNF should also be showed.
Why authors performed all the analysis after four weeks after transplantation? Different time points have been considered?
Transcranial injection is an invasive practice for human treatment. How this approach could clinically be used?
Minor
The reference n. 40 is the same as the reference n. 52
Author Response
In this manuscript Chen et al. investigated the effect of Sertoli cells on survival of the grafted dopaminergic neurons. They demonstrated that co-grafting of Sertoli cells and ventral mesencephalic tissue provided long-term beneficial effect to the grafts and improved the functional recovery in a rat model of Parkinson’s disease. They argued that these effects depended on immune-privileged microenvironment generated by Sertoli cells. This approach allows avoiding immunosuppressants and preventing the rejection.
This work is scientifically and well-written and presents interesting results although they lost originality. Different papers suggested the Sertoli transplantation as alternative treatment of neurodegenerative disorders such as Parkinson’s disease.
The research regarding the co-transplantation of dopaminergic neurons and Sertoli cells for the alternative treatment of neurodegeneration has been extensive.
Curiously, in a previous study by Chen et al, the allograft effect of Sertoli cells and rVM tissues in a PD rat model using [18F]FE-PE2I/ animal PET has been already evaluated. (J Nucl Med May 1, 2017 vol. 58 no. supplement 1 271). They already showed the effect of cotransplantation of rVM and Sertoli on tyrosine hydroxylase, dopamine transporter, T-cell and microglia markers, as in the present study.
Author response: We thank reviewer for this question. The publication (J Nucl Med May 1, 2017 vol. 58 no. supplement 1 271) was actually a poster abstract presented by our group at the symposium of Society of Nuclear Medicine and Molecular Imaging (SNMMI) in 2017. This poster contained some preliminary results of this manuscript. After the symposium, we worked on experiments of both allo- and xenotransplantation in the hemiparkinsonian rat model. All the data presented in this manuscript have not been published in any journal or in our knowledge by any other groups.
The release of neurotrophic factors such as GDNF/BDNF should also be showed.
Author response: We thank reviewer for this comment. According to a number of published studies, sertoli cells could release GDNF and BDNF (1-5). Therefore, we didn’t stain GDNF and BDNF in this study. The staining of neurotrophic factors can be conducted in future studies.
Why authors performed all the analysis after four weeks after transplantation? Different time points have been considered?
Author response: We thank reviewer for this comment. In this study, we focused on immunomodulatory effect of Sertoli cell co-grafted with ventral mesencephalic tissues. Based on the previous study, several immunological markers, such as CD4, MHC II (OX6) and CD11b were markedly increased around 3 weeks after transplantation of the ventral mesencephalic tissue but decreased 5 weeks post-transplantation (6). Therefore, we performed all the analyses 4 weeks after transplantation. In the future, we will prolong the experimental time and observe the long-term benefit of Sertoli cells co-grafted with ventral mesencephalic tissues in parkinsonian animal model.
Transcranial injection is an invasive practice for human treatment. How this approach could clinically be used?
Author response: We thank reviewer for this comment. Intrastriatal transplantations had been used in clinical applications (7, 8). In the recent study, two patients with Parkinson’s disease received intrastriatal transplantations of human fetal ventral mesencephalic tissue showed long-term motor improvement. Positron Emission Tomography image results also showed increased uptake of 18F-DOPA in striatum after the transplantation (9).
Minor
The reference n. 40 is the same as the reference n. 52
Author response: We thank the reviewer for pointing out these mistakes and suggestions, which have been corrected in the manuscript.
Reference:
Johnston DS, Olivas E, DiCandeloro P, Wright WW. Stage-Specific Changes in GDNF Expression by Rat Sertoli Cells: A Possible Regulator of the Replication and Differentiation of Stem Spermatogonia1. Biology of reproduction. 2011;85(4):763-9. Oliveira PF, Alves MG. Sertoli cell and germ cell differentiation. Sertoli Cell Metabolism and Spermatogenesis: Springer; 2015. p. 25-39. Hai Y, Hou J, Liu Y, Liu Y, Yang H, Li Z, et al. The roles and regulation of Sertoli cells in fate determinations of spermatogonial stem cells and spermatogenesis. Seminars in Cell & Developmental Biology. 2014;29:66-75. Li C, Zhou X. The potential roles of neurotrophins in male reproduction. 2013;145(4):R89. Meng X, Lindahl M, Hyvönen ME, Parvinen M, de Rooij DG, Hess MW, et al. Regulation of Cell Fate Decision of Undifferentiated Spermatogonia by GDNF. Science. 2000;287(5457):1489-93. Shinoda M, Hudson JL, Stro¨mberg I, Hoffer BJ, Moorhead JW, Olson L. Allogeneic grafts of fetal dopamine neurons: immunological reactions following active and adoptive immunizations. Brain Research. 1995;680(1):180-95. Peschanski M, Defer G, N'Guyen JP, Ricolfi F, Monfort JC, Remy P, et al. Bilateral motor improvement and alteration of L-dopa effect in two patients with Parkinson's disease following intrastriatal transplantation of foetal ventral mesencephalon. Brain. 1994;117(3):487-99. Spencer DD, Robbins RJ, Naftolin F, Marek KL, Vollmer T, Leranth C, et al. Unilateral Transplantation of Human Fetal Mesencephalic Tissue into the Caudate Nucleus of Patients with Parkinson's Disease. New England Journal of Medicine. 1992;327(22):1541-8. Kefalopoulou Z, Politis M, Piccini P, Mencacci N, Bhatia K, Jahanshahi M, et al. Long-term Clinical Outcome of Fetal Cell Transplantation for Parkinson Disease: Two Case Reports. JAMA Neurology. 2014;71(1):83-7.
Reviewer 2 Report
Manuscript ID: cells-626148
Title: The effect of Sertoli cells on xenotransplantation and allotransplantation of ventral mesencephalic tissue in a rat model of Parkinson’s disease
This is an interesting manuscript investigating the effect of Sertoli cells on trandplanted brain tissue in PD animal model through protection of death of dopaminergic neurons at xone- and allotransplantation. The authors have followed up subjected animals for 2 weeks after injection of cultured Sertoli cells. Then they have analyzed many kinds of molecular markers from lesional tissues of brain to demonstrate that the effects and the probable mechanism of neurologic outcome.
Although, this manuscript is well-organized based on a lots of scientific evidences, a number of issues dampen the reviewer's enthusiasm. There are major revisions which should be addressed before reconsidering this manuscript for publication.
The authors explain the benficial effects of therapeutic effect of Sertoli cells on structural and functional recovery relating to PD animal model. Although, they suggests some references, however, there are no clear therapeutic mechanisms of specific Sertoli cell action in transplated lesion of brain. Thus, more information and discussion need to be included to explain the differences of therapeutic mechanisms between general MCSs and Sertoli cells. While Sertoli cells are beneficial on various neural injuries, however it seems to be Sertoli cells have controversial effect, in particular induced benign or malignant neoplsm in neighbor or systemic active tissues. Therefore, to support authors‘s suggestion that Sertoli cells have neuroprtective and therapeutic effect in xeno- and allotransplanted brain lesion with PD symptom in this manuscript, it is required to add the evidence verifying neuronal recovery by implanted Sertoli cells from PD model (using neuronal marker such as NeuN, BDNF, and synaptogenesis markers as mandatory, etc.). There are explanation about stastistical significancy in each experimental groups for behavioral analysis, but not shown of how many animals subjected to each time points. Thus please provide clear statistical informations on this issue because it is critical results. Please thoroughly edit the manuscript for typographical and grammatical errors. For example, there are several instances where proper spacing is omitted. The authors also should thoroughly check in abbreviations used this manuscript.Author Response
This is an interesting manuscript investigating the effect of Sertoli cells on trandplanted brain tissue in PD animal model through protection of death of dopaminergic neurons at xone- and allotransplantation. The authors have followed up subjected animals for 2 weeks after injection of cultured Sertoli cells. Then they have analyzed many kinds of molecular markers from lesional tissues of brain to demonstrate that the effects and the probable mechanism of neurologic outcome.
Although, this manuscript is well-organized based on a lot of scientific evidences, a number of issues dampen the reviewer's enthusiasm. There are major revisions which should be addressed before reconsidering this manuscript for publication.
The authors explain the beneficial effects of therapeutic effect of Sertoli cells on structural and functional recovery relating to PD animal model. Although, they suggest some references, however, there are no clear therapeutic mechanisms of specific Sertoli cell action in transplanted lesion of brain. Thus, more information and discussion need to be included to explain the differences of therapeutic mechanisms between general MCSs and Sertoli cells.
Author response: Mesenchymal Stem cells (MSCs) is another promising cellular material for PD therapy. Similar to SCs, MSCs could secret neurotrophic factors, such as nerve growth factor (NGF), brain-derived neurotrophic factor (BDNF), and GDNF (1, 2). These factors could slow down degenerative progression of dopaminergic neurons and induce proliferation of neural stem cells (3). MSCs have been grafted bilaterally into striatum of early-stage PD patients, leading to improvement in Unified Parkinson's Disease Rating Scale (UPDRS) scores of the patients (4). These neuroprotective effects of MSCs may result from their anti-inflammatory properties. The effects of MSCs on lipopolysaccharides (LPS)-induced microglial activation have been evaluated in an in vitro study, indicating that MSCs could inhibit activation of microglia, reduce production of TNF-α and inducible nitric oxide synthase (iNOS), and increase production of anti-inflammatory cytokine IL-10 and transforming growth factor β (TGF-β) (5). These results suggest that MSCs might be used as a co-graft with the VM tissue in the cell therapy for late-stage PD patients. This is also added in the discussion of this manuscript.
SCs have been shown to secret numerous trophic factors, including insulin-like growth factor-I (IGF-I), basic fibroblast growth factor (bFGF), and glial cell line-derived neurotrophic factor (GDNF) (6). The beneficial effect of SCs observed in this study may be in part due to the release of GDNF, which enhances TH-positive cell survival and nerve fiber formation after transplantation of VM tissues in the striatum (7). Previous study showed that grafted dopaminergic precursor cells may not differentiate completely after transplantation (7), and may require several neurotrophic factors, such as GDNF and bFGF, to differentiate into mature cells (8). As mentioned above, SCs may be beneficial for the differentiation of dopaminergic neurons due to the release of GDNF and bFGF. This is included in second paragraph of discussion in this manuscript.
While Sertoli cells are beneficial on various neural injuries, however it seems to be Sertoli cells have controversial effect, in particular induced benign or malignant neoplasm in neighbor or systemic active tissues. Therefore, to support authors’ suggestion that Sertoli cells have neuroprotective and therapeutic effect in xeno- and allotransplanted brain lesion with PD symptom in this manuscript, it is required to add the evidence verifying neuronal recovery by implanted Sertoli cells from PD model (using neuronal marker such as NeuN, BDNF, and synaptogenesis markers as mandatory, etc.).
Author response: We thank reviewer for this comment. Sertoli cell tumor is a rare tumor in both animal and clinical studies (0.4–1.5% of all testicular neoplasms) (9, 10). We isolated Sertoli cells (SCs) from 3-week-old SD rat testis. It might not induce tumor in transplantation site. In this study, we specifically focus on dopaminergic neurons, so we stain tyrosine hydroxylase (TH) instead of NeuN. Based on the previous study, grafts of the ventral mesencephalic (VM) tissue in striatum of the hemiparkinsonian rats could not only establish a new dopaminergic terminal, but also form synaptic connection in the grafted site. TH-immunoreactive axons in grafted site striatum were seen to make abundant symmetric synapses (11). Furthermore, some neurotrophic factors, such as GDNF and BDNF have been shown released by SCs (12-16). BDNF could enhance functional reinnervation of the transplanted fetal dopamine neurons within striatum of the hemiparkinsonian rats (17). These published data provide several possible mechanisms and testable hypotheses that may explain the beneficial effect of Sertoli cells on the survival of VM allo- and xenografts and the behavioral recovery as well in the PD animal model. To further elucidate mechanism(s) underlying the SCs’ effect, however, would require a series of careful investigations, which might be conducted by our group in near future.
There are explanations about statistical significance in each experimental group for behavioral analysis, but not shown of how many animals subjected to each time points. Thus, please provide clear statistical informations on this issue because it is critical results.
Author response: We thank the reviewer for pointing out these mistakes and suggestions. The clear animals numbers has been provided of result in this manuscript.
Please thoroughly edit the manuscript for typographical and grammatical errors. For example, there are several instances where proper spacing is omitted. The authors also should thoroughly check in abbreviations used this manuscript.
Author response: We thank the reviewer for pointing out these mistakes and suggestions, which have been corrected in the manuscript.
Reference:
Garcı́a Ro, Aguiar J, Alberti E, de la Cuétara K, Pavón N. Bone marrow stromal cells produce nerve growth factor and glial cell line-derived neurotrophic factors. Biochemical and Biophysical Research Communications. 2004;316(3):753-4. Intravenous Administration of Marrow Stromal Cells (MSCs) Increases the Expression of Growth Factors in Rat Brain after Traumatic Brain Injury. 2004;21(1):33-9. Bonuccelli U, Del Dotto P. New pharmacologic horizons in the treatment of Parkinson disease. Neurology. 2006;67(7 suppl 2):S30-S8. Venkataramana N, Pal R, Rao SA, Naik AL, Jan M, Nair R, et al. Bilateral transplantation of allogenic adult human bone marrow-derived mesenchymal stem cells into the subventricular zone of Parkinson’s disease: a pilot clinical study. 2012;2012. Lee PH, Park HJ. Bone Marrow-Derived Mesenchymal Stem Cell Therapy as a Candidate Disease-Modifying Strategy in Parkinson's Disease and Multiple System Atrophy. J Clin Neurol. 2009;5(1):1-10. Willing AE, Othberg AI, Saporta S, Anton A, Sinibaldi S, Poulos SG, et al. Sertoli cells enhance the survival of co-transplanted dopamine neurons. Brain Research. 1999;822(1):246-50. Wakeman DR, Redmond DE, Dodiya HB, Sladek JR, Leranth C, Teng YD, et al. Human Neural Stem Cells Survive Long Term in the Midbrain of Dopamine-Depleted Monkeys After GDNF Overexpression and Project Neurites Toward an Appropriate Target. 2014;3(6):692-701. Yang F, Liu Y, Tu J, Wan J, Zhang J, Wu B, et al. Activated astrocytes enhance the dopaminergic differentiation of stem cells and promote brain repair through bFGF. Nature Communications. 2014;5:5627. Giglio M, Medica M, de Rose AF, Germinale F, Ravetti JL, Carmignani G. Testicular Sertoli Cell Tumours and Relative Sub-Types. Urologia Internationalis. 2003;70(3):205-10. Wakui S, Muto T, Kobayashi Y, Ishida K, Nakano M, Takahashi H, et al. Sertoli–Leydig Cell Tumor of the Testis in a Sprague-Dawley Rat. Journal of the American Association for Laboratory Animal Science. 2008;47(6):67-70. Freund T, Bolam J, Bjorklund A, Stenevi U, Dunnett S, Powell J, et al. Efferent synaptic connections of grafted dopaminergic neurons reinnervating the host neostriatum: a tyrosine hydroxylase immunocytochemical study. 1985;5(3):603-16. Johnston DS, Olivas E, DiCandeloro P, Wright WW. Stage-Specific Changes in GDNF Expression by Rat Sertoli Cells: A Possible Regulator of the Replication and Differentiation of Stem Spermatogonia1. Biology of reproduction. 2011;85(4):763-9. Oliveira PF, Alves MG. Sertoli cell and germ cell differentiation. Sertoli Cell Metabolism and Spermatogenesis: Springer; 2015. p. 25-39. Hai Y, Hou J, Liu Y, Liu Y, Yang H, Li Z, et al. The roles and regulation of Sertoli cells in fate determinations of spermatogonial stem cells and spermatogenesis. Seminars in Cell & Developmental Biology. 2014;29:66-75. Li C, Zhou X. The potential roles of neurotrophins in male reproduction. 2013;145(4):R89. Meng X, Lindahl M, Hyvönen ME, Parvinen M, de Rooij DG, Hess MW, et al. Regulation of Cell Fate Decision of Undifferentiated Spermatogonia by GDNF. Science. 2000;287(5457):1489-93. Yurek DM, Lu W, Hipkens S, Wiegand SJ. BDNF Enhances the Functional Reinnervation of the Striatum by Grafted Fetal Dopamine Neurons. Experimental neurology. 1996;137(1):105-18.Round 2
Reviewer 1 Report
The authors have addressed the comments I made.